# Specific c-Jun N-Terminal Kinase Inhibitor, JNK-IN-8 Suppresses Mesenchymal Profile of PTX-Resistant MCF-7 Cells through Modulating PI3K/Akt, MAPK and Wnt Signaling Pathways

**DOI:** 10.3390/biology9100320

**Published:** 2020-10-01

**Authors:** Pelin Ozfiliz Kilbas, Ozlem Sonmez, Pinar Uysal-Onganer, Ajda Coker Gurkan, Pinar Obakan Yerlikaya, Elif Damla Arisan

**Affiliations:** 1Department of Molecular Biology and Genetics, Istanbul Kultur University, 34158 Istanbul, Turkey; p.ozfiliz@iku.edu.tr (P.O.K.); sonmezzozlem@gmail.com (O.S.); a.coker@iku.edu.tr (A.C.G.); p.obakan@iku.edu.tr (P.O.Y.); 2Cancer Research Group, School of Life Sciences, University of Westminster, London W1W 6UW, UK; P.onganer@westminster.ac.uk; 3Institute of Biotechnology, Gebze Technical University, 41400 Kocaeli, Turkey

**Keywords:** Paclitaxel, JNK, WNT, drug resistance, breast cancer

## Abstract

**Simple Summary:**

Investigation into effective targets of drug resistance is important for identifying novel strategies in cancer therapy. The study aimed to determine the functional role of paclitaxel (PTX) resistance on MCF-7 cell survival related to PI3K/Akt and MAPK pathways. Therefore, we generated PTX-resistant (PTX-res) MCF-7 cells exposed to increasing concentrations of PTX (5–100 nM) over a period of 6 months. Increased cell survival, proliferation, and colony formations were observed in PTX-res MCF-7 cells, while survival inhibition was determined in non-resistant wt cells. PTX-res MCF-7 cells appeared morphologically different from wt cells with their star-like shape which showed the mesenchymal characteristics of cells. Active PI3K/Akt signaling and increased motility were confirmed by upregulation of the EMT pathway members in PTX-res MCF-7 cells. We suggested that the active Akt signaling was related to the upregulated stress-mediated activation of MAPK signaling members, as shown by the significant p38 and SAPK/JNK activation in our results. To sensitize PTX-res MCF-7 cells we treated wt and PTX-res MCF-7 cells with specific c-Jun N-terminal kinase inhibitor, JNK-IN-8, and significant suppression on p38, SAPK/JNK expression was observed. Wnt signaling was highly affected by JNK inhibition. We concluded that JNK inhibition is a potential target to reverse PTX-resistance related to Wnt signaling.

**Abstract:**

Paclitaxel (PTX) is a widely used chemotherapeutic agent in the treatment of breast cancer, and resistance to PTX is a common failure of breast cancer therapy. Therefore, understanding the effective molecular targets in PTX-resistance gains importance in identifying novel strategies in successful breast cancer therapy approaches. The aim of the study was to investigate the functional role of PTX resistance on MCF-7 cell survival and proliferation related to PI3K/Akt and MAPK pathways. The generated PTX-resistant (PTX-res) MCF-7 cells showed enhanced cell survival, proliferation, and colony formation potential with decreased cell death compared to wt MCF-7 cells. PTX-res MCF-7 cells exhibited increased motility profile with EMT, PI3K/Akt, and MAPK pathway induction. According to the significant SAPK/JNK activation in PTX-res MCF-7 cells, specific c-Jun N-terminal kinase inhibitor, JNK-IN-8 is shown to suppress the migration potential of cells. Treatment of JNK inhibitor suppressed the p38 and SAPK/JNK and Vimentin expression. However, the JNK inhibitor further downregulated Wnt signaling members in PTX-res MCF-7 cells. Therefore, the JNK inhibitor JNK-IN-8 might be used as a potential therapy model to reverse PTX-resistance related to Wnt signaling.

## 1. Introduction

Breast cancer is one of the most common malignancies among women, impacting 2.1 million women each year, and it also causes the greatest number of cancer-related deaths. In 2018, it was estimated that 627,000 women died from breast cancer, which was approximately 15% of all cancer deaths [1]. Among the current therapy approaches, chemotherapy is one of the major treatments for breast cancer [2]. However, intrinsic or acquired resistance to chemotherapeutic drugs leads to the failure of chemotherapy in many patients with breast cancer [3]. Therefore, it is critical to explore the molecular mechanisms of drug resistance in breast cancer and find novel therapeutic targets to overcome drug resistance and improve the survival of breast cancer patients [4]. Drug resistance is a well-known phenomenon that results in unresponsiveness to chemotherapy through increased drug efflux, metabolic changes that promote drug inhibition and degradation, and DNA mutations [5]. Taxol (Paclitaxel), a promising therapeutic agent in the treatment of metastatic breast cancer, is isolated from *Taxus brevifolia*, the Pacific yew tree, and promotes the microtubule polymerization and stabilization of living cells [6]. It is a frequently used chemotherapy agent for breast cancer treatment, therefore, resistance to paclitaxel is a common limiting factor for breast cancer therapies. Drug resistance may be intrinsic, making malignant cells resistant to numerous unrelated drugs or may be acquired after exposure to chemotherapy [7]. PTX resistance in several cancer cells such as breast and ovarian was associated with elevated levels of multidrug resistance (MDR) protein. MDR1 protein, which is also named P-glycoprotein or ABCB1, protects cells from chemical toxicity and oxidative stress leading to cell survival [8]. MDR is related to various survival targeted pathways such as phosphatidylinositol-3-kinase (PI3K) /Akt, mitogen-activated protein kinase (MAPK), and epithelial–mesenchymal transition (EMT) [9,10]. 

The PI3K/Akt signaling pathway plays an important role in regulating cell proliferation and preserving the biological properties of malignant cells [11]. PI3K is a heterodimer consisting of a regulatory subunit p85 and a catalytic subunit p110 and activated with tyrosine kinase to form PIP3 in the plasma membrane. PIP3 then interacts with the PH domain to cause Akt to accumulate in the membrane. Then, 3-phosphoinositide-dependent protein kinase 1 (PDK1) phosphorylates and activates the Thr308 residue of Akt protein kinase. In addition, the inhibition of mTORC1 can induce Akt Ser473 phosphorylation in a subset of cancer cell lines and patient tumors [12]. The activated Akt then phosphorylates a number of substrates, thereby affecting various cellular and physiological processes such as cell cycle, cellular growth, differentiation, survival, apoptosis, angiogenesis, migration and EMT [13,14].

EMT pathway plays a central role in invasion and metastasis, where epithelial cells lose stable cell–cell adhesion and are rearranged to achieve the migration activity of the cytoskeleton. In EMT conditions, the epithelial marker E-cadherin is separated from the plasma membrane and then degraded, and the loss of E-Cadherin is balanced by increasing the expression of the mesenchymal neural cadherin (N-cadherin) and results in the increase in cell migration and invasion [15]. Changes in gene expression that contribute to the suppression of the epithelial phenotype and the activation of the mesenchymal phenotype include Snail, Twist, Zinc-finger box-binding (ZEB) transcription factors. They usually control each other’s expression and functionally cooperate with target genes and trigger EMT progression [16]. As a result of EMT, tumor cells show characteristic features of cancer stem cells with superior tumorigenic effects and develop more powerful drug-resistant properties that promote tumor recurrence and metastasis [17]. 

Wnt signaling is an evolutionarily conserved pathway that regulates cell migration and proliferation and cancer progression [18]. Wnt signaling pathway is divided into two branches: the canonical β-catenin-dependent pathway and the non-canonical planar cell polarity (PCP) and Wnt/Ca2+ pathways. Wnt proteins presented on the surface of signaling cells act on target cells by binding to the Frizzled (Fz)/low density lipoprotein receptor-related protein (LRP) complex at the cell surface. When the Wnt proteins bind to the membrane-bound spiked receptors and correlators to the LRP 5 and 6, a number of the signal cascade is activated [19]. These receptors transduce a signal to several intracellular proteins that include Dishevelled (Dsh), glycogen synthase kinase-3β (GSK-3β), Axin, Adenomatous Polyposis Coli (APC), and the transcriptional regulator, β-catenin. Cytoplasmic β-catenin levels are normally kept low through continuous proteasome-mediated degradation, which is controlled by a complex containing GSK-3β/APC/Axin. When cells receive Wnt signals, the degradation pathway is inhibited, and consequently, β-catenin accumulates in the cytoplasm and nucleus. Nuclear β-catenin interacts with transcription factors such as the lymphoid enhancer-binding factor 1 (LEF1)/T cell-specific transcription factor (TCF) to affect the transcription of EMT targeted genes [20]. The noncanonical pathway, which is independent of β-catenin, mediates several cellular processes through monomeric GTPases of the Rho family, mitogen-activated protein kinase-like Jun N-terminal kinase (JNK), and changes in intracellular calcium levels [21,22]. Wnt signals are important for the EMT of cancer cells, which promotes cancer invasion and metastasis. 

The JNK protein family is key transducers of extracellular stress signals and can provide a therapeutic strategy for the treatment of a variety of diseases such as neurodegeneration, cancer, and autoimmune diseases. JNK-IN-8 is one of the first irreversible c-Jun N-terminal kinase JNK inhibitor. JNK-IN-8, which was highly selectively bound to JNK and inhibited this protein, was used to sensitize aggressive breast cancer cells to chemotherapeutic drugs [23].

In this study, we aim to investigate the potential role of PTX-resistance on cell survival, proliferation, and the growth potential of MCF-7 cells. The PTX-resistant MCF-7 cells showed increased survival properties due to enhanced Akt and MAPK activation that followed by the transition of cells from epithelial to mesenchymal characteristics. The enhanced EMT function of cells triggered migration of PTX-res MCF-7 cells compared to wild type (wt) cells. A JNK-IN-8 inhibitor was used to suppress these migratory functions of cells through inhibiting p38, SAPK/JNK, and Wnt signaling members. Therefore, JNK-IN-8 might be a potential therapeutic target to reverse the PTX-resistance of MCF-7 breast cancer cells.

## 2. Materials and Methods

### 2.1. Chemicals and Antibodies

Paclitaxel (PTX) was purchased from Bristol-Myers Squibb, dissolved in Ethanol to make 7 μM stock concentration. JNK-IN-8 (Calbiochem, UK), with a working concentration of 10 µM was prepared in DMSO. MDR1/ABCB1, E-cadherin, N-cadherin, Vimentin, Snail, TCF8/ZEB1, β-catenin, Axin-1, Claudin, PI3K p110γ, p-PDK-1, Akt, p-Akt, p-c-Raf, Cyclin D1, p-GSK3β, p-p38 MAPK, p38 MAPK, p-p44/42, c-jun, p- SAPK/JNK, SAPK/JNK, p-AMPK, Wnt3a, Wnt5a/b, p-LRP6, LRP6, Dvl2, Dvl3, β-actin, GAPDH primary, and secondary antibodies were used at 1:1000 dilutions and were from Cell Signaling Technology (CST, Beverly, MA, USA). 

### 2.2. Cell Culture 

MCF-7 (HTB-22) human breast cancer cells (ATCC, Manassas, VA, USA) and generated paaclitaxel-resistant (PTX-res) MCF-7 cells were cultured in Dulbecco’s Modified Eagle’s Medium (DMEM) (Gibco-Life Technologies, Grand Island, NY, USA) supplemented with 10% fetal bovine serum (Pan Biotech, Germany) and antibiotics (10000 U penicillin/mL, 10 mg streptomycin/mL (Pan Biotech, Aidenbach, Bavaria, Germany). Both cell lines were maintained in a standard humidified 5% CO_2_ incubator at 37 °C (Hera Cell 150i, Thermo Scientific, Waltham, MA, USA). 

MCF-7 breast cancer wild type cells were exposed to increasing concentrations of PTX (5–100 nM) over a period of 6 months to generate repetitive clones of PTX-res cells against 100 nM PTX treatment. The generation of PTX-resistant MCF-7 cells were performed in at least three different experiments as different batch clones. In order to prevent the loss of PTX resistant colonies, we prepared six different batch clones and utilized at least three of them in confirmation experiments. Cells were treated with PTX for 24 h. Then, cell debris was removed and the remaining cell population was maintained in fresh media for at least 1 week. The increased PTX concentration was shown in Figure 1A. 

### 2.3. Trypan Blue Dye Exclusion Assay

The wt and PTX-res MCF-7 cells were seeded at 1 × 10^5^ density in 6-well plates (TPP, Zollstrasse, Trasadingen, Switzerland) and treated with 100 nM PTX within 72 h. First, cells were trypsinized (Trypsin EDTA (0.25%), Gibco, USA), and centrifuged. Then, cells were exposed to 0.4% (*w*/*v*) Trypan Blue (Gibco, USA) (50 µL) and DMEM (50 µL) at a 1:1 ratio. After that, 10 µL of cells were counted by a dual-chamber 0.1 mm deep Neubauer improved hemocytometer. Viable and trypan blue-stained non-viable cells were recorded and, based on viable cells, a graph was formed. The obtained data were plotted on a graph for the cell number (y-axis) and time (x-axis).

### 2.4. Colony Formation Assay

The wt and PTX-res MCF-7 cells were seeded at 1 × 10^4^ density into 6-well plates. Following their attachment, both were treated with 100 nM PTX for 14 days. Then, the media was removed and cells were washed with 1×-PBS solution, fixed with methanol: acetic acid (3:1) for 5 min. Following the removal of fixing agents, cells were stained with 0.5% crystal violet in methanol for 15 min, washed by distilled water, and the morphological images were taken under light microscopy. Scale bar is 10 µm and magnification is 20×. 

### 2.5. Soft Agar Assay

The base agar was prepared to create a mixture with equal amounts of 2× DMEM medium (20% FBS and 2% penicillin/streptomycin) and 0.5% agarose (Sigma Aldrich, St. Louis, MO, USA) in 1× PBS and the mixture was dispersed as 1 mL into each well in 6-well plates. Once the lower layer of agar had solidified, the upper layer was prepared. Then, the induration of gel 3.0 × 10^5^ cells/mL in a 1:1 mix of 2× DMEM medium and 0.3% agarose solution, and the solution was placed on the solidified agar. After solidification, samples were kept at 37 °C for 15–18 days. Colony formation potentials were examined by light microscopy (Olympus IX71, Shinjuku-ku, Tokyo, Japan). The diameter of the spheroid formation was analyzed by using the Olympus Micro DP Manager Image Analysis program in a time-dependent manner.

### 2.6. Fluorescence Microscopy

#### 2.6.1. Propidium Iodide (PI) Staining

The wt and PTX-res MCF-7 cells were seeded at 3 × 10^4^ cells per 6-well plates and treated with 100 nM PTX for 24 h. Through drug treatment, cells were washed with 1×-PBS, and PI was performed to cells (50 mg/mL stock concentration in 1×-PBS) and incubated for 30 min in the incubator. Dead cells (red-stained cells), which occurred through drug treatment, were determined by fluorescence microscopy (Olympus, Japan) in excitation 535 nm and emission 617 nm.

#### 2.6.2. 3,3′-Dihexyloxacarbocyanine Iodide (DiOC6) Staining

Cells were seeded at 3 × 10^4^ density into 6-well plates and treated with 100 nM PTX for 24 h. Then, cells were washed with 1×-PBS and stained with 4nM DiOC6 (40 nM stock concentration in DMSO (Calbiochem, USA) for 15 min in the dark. Mitochondrial membrane potential (MMP) disruption was determined by fluorescence microscopy (Olympus, Japan) in excitation 482 and emission 504 nm.

### 2.7. Protein Extraction and Immunoblotting

The wt and PTX-res MCF-7 cells were treated with 100 nM PTX and/or 10 μM JNK-IN-8 for 24 h. Then, cells were washed with ice-cold 1×-PBS and lysed in M-PER Mammalian Protein Extraction Reagent (Thermo Scientific, USA) with a protease inhibitor cocktail (Roche, Indianapolis, IN, USA). After lysis, cells were centrifuged for 20 min at 16,000 g and total protein concentrations were determined with Bradford protein assay (Bio-Rad, Hercules, CA, USA). Then, 30 µg total protein was separated into 12% sodium dodecyl sulfate-polyacrylamide gels (SDS-PAGE) and transferred to polyvinylidene difluoride (PVDF) membranes (Thermo Scientific, USA). Following the washing of membranes in Tris-buffered Saline with Tween-20 (TBS-T) (10mM Tris-HCl pH.8, 0.05% Tween-20), PVDF membranes were blocked by 5% skim milk containing TBS-T for 1h at room temperature. Then, PVDF membranes were incubated in primary antibody buffer containing 5% (v/v) skim milk solution (MDR1/ABCB1) (D3H1Q) (1:1000), E-cadherin (24E10) (1:1000), N-cadherin (D4RH1) (1:750), Vimentin (D21H3) (1:1000), Snail (C15D3) (1:1000), TCF8/ZEB1 (D80D3) (1:1000), β-catenin (D10A8) (1:1000), Axin-1 (C7B12) (1:500), Claudin (D5H1D) (1:750), PI3K p110γ (D55D5) (1:1000), p-PDK-1 Ser 241 (C49H2) (1:1000), Akt (C67E7) (1:1000), p-Akt Ser473 (D9E) (1:1000), p-c-Raf Ser259 (1:1000), Cyclin D1 (92G2)(1:1000), p-GSK-3β (1:1000), p-p38 MAPK Thr180/Tyr182 (D3F9) (1:1000), p38 MAPK (1:1000), p-p44/42 Thr202/Tyr204 (D13.14.4E) (1:1000), c-Jun (60A8) (1:1000), p- SAPK/JNK Thr183/Thy185 (81E11) (1:1000), SAPK/JNK (1:1000), p-AMPK Thr172 (40H9) (1:1000), Wnt3a (C64F2) (1:1000), Wnt5a/b (C27E8) (1:1000), p-LRP6 Ser1490 (1:1000), LRP6 (C5C7) (1:1000), Dvl2 (30D2) (1:1000), Dvl3 (1:1000), β-actin (13E5) (1:1000), GAPDH (14C10) (1:1000) (Cell-Signaling Technology, USA) overnight at 4 °C. Then membranes were rinsed with TBS-T and incubated with horseradish peroxidase (HRP)-conjugated secondary antibodies (anti-rabbit IgG or anti-mouse IgG (Cell-Signaling Technology, USA) secondary antibodies for overnight at 4 °C. Following the addition of enhanced chemiluminescence reagent, signals from the HRP-coupled antibodies were detected using the ChemiDoc MP Imaging System (Bio-Rad Laboratories, Hercules, CA, USA). All proteins were quantified relative to the loading control β-actin and GAPDH.

### 2.8. Wound-Healing Assay

The wt and PTX-res MCF-7 cells were seeded at 1 × 10^6^ cells and grown to 80% confluence in 35 mm plates, and the cell monolayer was then scratched with the narrow end of a sterile 200 µL pipette tip. Then, the medium was promptly replaced to eliminate floating cells and exchanged with DMEM. The width of the scratch was measured at two points in each well after initial wounding. The cells were incubated for 24 h at 37 °C in a CO_2_ incubator, and then, the scratch width was re measured. The relative motility and migration ability of the cells into the cell-free zone is expressed as the normalized percent change in the scratch width after 24, 48 and 72 h.

### 2.9. RNA Isolation, cDNA Synthesis, and RT-PCR

RNA isolation from wt and PTX-res MCF-7 cells were performed using TRIPure (Roche, USA) according to the manufacturer’s indications. cDNA was synthesized from total by using iScript cDNA Synthesis Kit (Bio-Rad, USA). Twist, 18S rRNA genes were amplified using synthesized cDNA by RT-PCR (Mini Thermal Cycler, Bio-Rad, USA). Then, electrophoresis was performed at 1.5% agarose gel, visualized by Gel Detection System (Bio-Rad, USA).

### 2.10. MTT Cell Viability Assay

The effects of PTX and JNK-IN-8 on cell viability were determined by colorimetric 3-(4,5-dimethylthiazol-2-yl) 2,5-diphenyl-tetrazolium bromide (MTT; Roche, USA) assay. Cells were seeded at a density of 8 × 10^3^ cells/well in 96-well plates, allowed to attach overnight and treated with 10 μM JNK-IN-8 and/or 100 nM PTX for 24 h. After drug treatment, 10 μL of MTT reagent (5 mg/mL) was added to the cell culture medium for 4 h. Following the removal of media, 200 μL DMSO was added to dissolve the formazan crystals, which are produced due to activated mitochondria. The absorbance of the suspensions was determined at 570 nm with a microplate reader (Bio-Rad, USA).

### 2.11. Statistical Analysis

All the statistical analysis of experiments was performed using GraphPad Prism version 8.0.1, Available online: https://www.graphpad.com/ (GraphPad Software, San Diego, CA, USA, accessed on 20 August 2020). MTT cell viability assay, Trypan blue dye exclusion assay, Colony formation assay, Soft Agar Assay, and Wound Healing Assay were repeated three times and statistically analyzed using two-way ANOVA Sidak’s Multiple comparison test. The densitometric calculation of three replicated immunoblotting images was performed using Image Lab Software (Bio-Rad, USA) and analyzed by Two-Way ANOVA, Tukey’s multiple comparison test. Statistically significant values considered as following: * *p* < 0.05; ** *p* < 0.001; *** *p* < 0.001; **** *p* < 0.0001. Error bars represent ± standard deviation values. 

## 3. Results

### 3.1. Establishment and Determination of Drug Resistance of PTX-Res MCF-7 Breast Cancer Cell Line

PTX-res MCF-7 cells were generated by treating the cells with increased PTX concentrations for 6 months. First, MCF-7 cells were treated with PTX 5,10 and 20 nM for 24 h, and then PTX concentration was increased gradually. The overview of the resistance strategy was shown in Figure 1A. Following 100 nM PTX treatment, the live colonies were selected and names as PTX-res MCF-7 cells for further experiments. The morphology of the cells was observed and noted that the PTX-res MCF-7 cells formed an elongated and polarized shape compared to round-like wt cells. To determine the PTX resistance phenotype, wt, and PTX-res MCF-7 cells were treated with 100 nM PTX for 24 h, and the expression profile of membrane-associated, drug-resistant protein MDR/ABCB1 was investigated by immunoblotting assay. While MDR/ABCB1 expression was not observed in wt cells, remarkable upregulation of MDR/ABCB1 was observed in both untreated and PTX treated MCF-7 PTX-res cell without significant alteration between them (n = 3, **** *p* < 0.0001) (Figure 1B). β-tubulin was selected as a loading control. 

### 3.2. PTX-Resistance Enhanced the Proliferation and Colony Formation Potential of MCF-7 Cells

To determine the potential effect of PTX-resistance on MCF-7 cells, we performed trypan blue dye exclusion cell proliferation, colony formation, and soft agar assays. The proliferation ratios of wt and PTX-res MCF-7 cells were determined in time-dependent (0–72 h) PTX treatment. Our results showed that the viable cell number of PTX-res MCF-7 cells was significantly higher than wt cells in each time condition (n = 3, **** *p* < 0.0001). The treatment of wt MCF-7 cells with 100 nM PTX for 24 h decreased the viable cell number, but the proliferation ratio of wt cells slightly increased within 48 and 72 h treatment (n = 3, **** *p* < 0.0001) compared to untreated wt cells. PTX treatment did not cause significant cell number loss in PTX-res MCF-7 cells (Figure 2A). Similarly, PTX treatment decreased the colony-forming potential of wt MCF-7 cells, however, PTX-res increased the number of colonies in both untreated and PTX-treated MCF-7 cells compared to wt cells (n = 3, **** *p* < 0.0001). PTX treatment did not cause a significant colony number decrease in PTX-res MCF-7 cells (Figure 2B). Concomitantly, PTX-res MCF-7 cells showed the increased anchorage-independent cell proliferation and 3D sizes of colonies compared to wt cells both on Day 7 and Day 18. Although PTX treatment significantly decreased the colony sizes of wt cells, PTX-res MCF-7 cells did not show a remarkable colony size decrease upon PTX treatment both in Day 7 and Day 18 (Figure 2C). The effect of PTX on cell death and mitochondrial membrane potential in wt and PTX-res MCF-7 was demonstrated by PI and DiOC6 staining. It was observed that PTX treatment increased the PI-positive dead cell number (red-fluorescent cells) in wt MCF-7 cells compared to untreated wt MCF-7 cells. However, PTX-res MCF-7 cells did not show a significant cell death number upon PTX treatment. DiOC6 staining, which is regarded as the mitochondrial membrane potential marker, diminished upon PTX treatment in wt MCF-7 cells, but not in both untreated and PTX-treated PTX-res MCF-7 cells (Figure 2D). 

### 3.3. PTX-Resistance Modulated the PI3K/Akt and MAPK Pathways 

The biological function of PTX-resistance on PI3K/Akt and MAPK pathways, which are important molecular signaling routes associated with cell survival, was examined by immunoblotting. PI3K expression was significantly higher in both untreated and PTX-treated PTX-res MCF-7 cells compared to wt MCF-7 cells (n = 3, **** *p* < 0.0001). Phospho-PDK1 at S241 residue and total Akt did not show a significant expression change in untreated and PTX-treated wt and PTX-res MCF-7 cells, whereas phospho-Akt at S473 residue was remarkably downregulated in PTX-treated wt cells (n = 2, **** *p* < 0.0001). The inhibitory phosphorylation of GSK3β at S9 was significantly downregulated in PTX-treated wt cells but upregulated in PTX-treated PTX-res cells (n = 2, **** *p* < 0.0001). Cyclin D1 expression decreased with the PTX-treatment in wt cells, whereas it slightly increased both in untreated and PTX-treated PTX-res MCF-7 cells (n = 2, **** *p* < 0.0001). GAPDH was used as a loading control (Figure 3A). Densitometry analysis was shown in Appendix A. 

We then investigated the expression profiles of MAPK pathway members associated with the effect of PTX-resistance in MCF-7 cells; Phospho-c-Raf at S259 showed a slight upregulation in PTX-treated PTX-res cells compared to untreated PTX-res MCF-7 cells (Figure 3B; n = 2, ** *p* = 0.0049). Total p38 expression slightly increased in both untreated and treated PTX-res MCF-7 cells compared to wt cells (n = 2, ** *p* = 0.0060). Besides, the remarkable upregulation was shown in the expression profile of phospho-p38 at T180/Y182 in PTX-res cells compared to wt cells (n = 2, **** *p* < 0.0001, ** *p* = 0.0013), and the expression was higher in untreated PTX-res cells compared with PTX-treated PTX-res cells (n = 2, ** *p* = 0.0010). Similarly, total p44/42 expression was increased in both untreated and PTX-treated MCF-7 cells compared to untreated and PTX-treated wt cells (n = 2, * = 0.0381, ** *p* = 0.0016). phospho-p44/42 expression was slightly downregulated in PTX-treated wt cells while it increased both in untreated and PTX-treated PTX-res MCF-7 cells (n = 2, * = 0.0130, ** *p* = 0.0016). c-Jun expression was slightly downregulated in both untreated and PTX-treated PTX-resistant MCF-7 cells compared to wt cells (n = 2, ** *p* = 0.0023). It was observed that phospho-SAPK/JNK at T183/Y185 was upregulated in untreated PTX-res cells compared to untreated wt cells (n = 2, ** *p* = 0.0013). The treatment of PTX downregulated phospho-SAPK/JNK in wt and PTX-res MCF-7 cells (n = 2, *** *p* = 0.0002). The expression profile of phospho-AMPK-α at T172 as another Akt pathway mediator was significantly upregulated in both untreated and PTX-treated PTX-resistant MCF-7 cells compared to wt cells (n = 2, **** *p* < 0.0001). Densitometry analysis was shown in Appendix A. 

### 3.4. PTX-Resistance Enhance the Migratory Potential of MCF-7 Cells

We investigated the expression profiles of EMT pathway members, which are associated with cell metastasis and drug resistance mechanisms. The expression of E-cadherin as an epithelial marker increased in both untreated and treated wt cells compared to PTX-res cells (n = 3, **** *p* < 0.0001). Similarly, Claudin, another epithelial marker, showed a downregulated expression profile in PTX-res MCF-7 cells compared to wt cells (n = 2, **** *p* < 0.0001), but PTX-treatment slightly enhanced its expression in PTX-res MCF-7 cells (n = 2, ** *p* = 0.0013). The expression levels of mesenchymal markers N-cadherin and Vimentin were remarkably increased in PTX-res MCF-7 cells compared to wt cells (n = 3, **** *p* < 0.0001). The transcription factor of EMT signaling marker Snail showed a diminished expression in both untreated and PTX-treated wt cells and was slightly upregulated in PTX-res cells upon PTX treatment (n = 2, ** *p* = 0.0084, *** *p* = 0.0007). Another transcription factor of the EMT pathway, TCF8/ZEB1 expression, was slightly upregulated in wt cells upon PTX treatment. In addition, PTX-resistant significantly enhanced its expression both in untreated and PTX-treated PTX-res MCF-7 cells compared to wt cells (n = 2, *** *p* = 0.0003). The expression level of β-catenin did not show a significant change in wt and PTX-res cells upon PTX treatment. The β-catenin-mediated Twist mRNA expression was increased through PTX resistance in MCF-7 cells, as shown in Appendix A. Axin-1 expression was enhanced in both untreated and PTX-treated MCF-7 cells upon PTX-resistance compared to wt cells (Figure 4A; n = 2, **** *p* < 0.0001). Densitometry analyses are shown in Appendix A.

We examined the wound closure potentials of wt and PTX-res MCF-7 cells with 100 nM PTX treatment to determine the motility characteristics of cells (Figure 4B). The wound distance was significantly decreased in untreated wt and PTX-res MCF-7 cells in a time-dependent manner, however, PTX-resistance was more effective on the motility of MCF-7 cells into the wound area. Although it was observed that PTX-treatment abrogated the wound closure capacity of wt MCF-7 cells, PTX-res MCF-7 cells were still providing wound healing in time-dependent PTX treatment.

### 3.5. JNK Inhibitor Suppressed Stress-Activated p38 MAPK and SAPK/JNK, and Mesenchymal Marker Vimentin Expression

According to our results that the stress-activated MAPK signaling members were significantly upregulated in PTX-res MCF-7 cells, we investigated the effect of a specific c-Jun N-terminal kinase JNK inhibitor on MAPK, EMT, Wnt signaling members associated with PTX-res mechanism (Figure 5). First, we determined the effect of individual treatments of PTX or JNK inhibitors and combinational PTX and JNK inhibitor treatment on the cell viability of wt and PTX-res MCF-7 cells. The cell viability results of wt and PTX-res MCF-7 cells confirmed that individual treatment of PTX on wt MCF-7 cells decreased the cell viability by 15% in wt cells (n = 3, ** *p* = 0.0054) but did not cause any significant cell viability decrease in PTX-res MCF-7 cells. However, only JNK inhibitor treatment (10 μM) was more effective than PTX on the cell survival of both wt and PTX-res MCF-7 cells. wt MCF-7 cells showed 76% and PTX-res MCF-7 cells showed 61% cell viability upon only JNK inhibitor treatment. It was shown that JNK inhibitor treatment caused a greater cell viability decrease in PTX res MCF-7 cells compared to wt cells (n = 3, ** *p* = 0.0059). Combinational treatment of PTX and JNK inhibitor treatment showed similar results to individual JNK inhibitor treatment (Figure 5A; n = 3, * *p* = 0.0147).

Although PTX treatment downregulated p38 expression in wt cells, slightly upregulated in both untreated and PTX-treated PTX-res MCF-7 cells compared to wt MCF-7 cells (-wt vs—PTX-res n = 2, ** *p* = 0.0036, PTX-treated wt vs—PTX-treated PTX-res n = 2, * *p* = 0.0498). Individual treatment of JNK inhibitor downregulated p38 expression in wt cells compared to untreated wt cells (-wt vs—JNK inhibitor-treated wt n = 2, *** *p* = 0.0004), however, combinational treatment of PTX and JNK inhibitor showed a similar expression profile to only PTX-treatment. JNK inhibitor treatment in both untreated and PTX-treated PTX-res MCF-7 cells upregulated p38 expression compared to JNK-inhibitor-treated wt cells. The expression profile of phospho-p38-MAPK at Thr180/Tyr182 residue increased in both untreated and PTX-treated PTX-res MCF-7 cells compared to wt cells (-wt vs—PTX-res n = 2, ** *p* = 0.0079, PTX-treated wt vs—PTX-treated PTX-res ** *p* = 0.0038). JNK inhibitor significantly decreased the phospho-p38-MAPK at Thr180/Tyr182 in wt and PTX-res MCF-7 cells. Total SAPK/JNK expression was significantly downregulated upon PTX treatment both in wt and PTX-res MCF-7 cells (-wt vs—PTX-res n = 3, ** *p* = 0.0027, PTX-treated wt vs—PTX-treated PTX-res **** *p* < 0.0001). However, the treatment of JNK inhibitor abolished SAPK/JNK expression in both untreated and PTX-treated wt and PTX-res MCF-7 cells. Following the investigation of the suppressive effect of JNK-inhibitor on MAPK pathway members, we showed the expression profile of E-cadherin and Vimentin as EMT pathway regulators. Although the expression profile of epithelial marker E-cadherin was not observed in PTX-res MCF-7 cells, combined treatment of JNK inhibitor and PTX treatment significantly upregulated E-cadherin expression in wt MCF-7 cells compared to individual JNK inhibitor treatment (JNK inhibitor-treated wt vs— JNK inhibitor and PTX-treated wt cells n = 3, *** *p* = 0.0001). In contrast, Vimentin expression was not observed in wt cells and its expression profile was significantly downregulated in only JNK inhibitor and combined JNK inhibitor and PTX-treated PTX-res cells compared to untreated and only-PTX-treated PTX-res MCF-7 cells (n = 2, **** *p* < 0.0001). The expression profile of MDR/ABCB1 did not show a significant change upon JNK inhibitor treatment in PTX-res cells compared to only PTX-treated PTX-res cells (Figure 5B). Densitometry analysis was shown in Appendix A.

### 3.6. JNK Inhibitor Suppressed the Expression Profiles of Wnt Signaling Proteins

To understand the effect of JNK inhibitor on cell migration and invasion, we determined the the expression profiles of Wnt signaling members. Wnt3a protein expression level was increased in both untreated and PTX-treated PTX-res cells MCF-7 cells compared to wt cells (-wt vs—PTX-res n = 2, ** *p* = 0.0010, PTX-treated wt vs— PTX-treated PTX-res *** *p* = 0.0001). Treatment of only the JNK inhibitor and combined treatment with PTX downregulated Wnt3a expression in wt and PTX-res MCF-7 cells. A similar expression profile was observed in the Wn5a/b expression. The downstream members of the Wnt signaling pathway, phospho- LRP6 at Ser1490, Dvl2 and Dvl3 and Axin-1 showed similar expression profiles, as they were upregulated in both PTX untreated/treated PTX-res cells compared to wt cells, however only the JNK inhibitor and combined treatment of JNK inhibitor significantly decreased this upregulation (Figure 6). Densitometry analysis was shown in Appendix A.

## 4. Discussion

Treatment of breast cancer patients with a classical chemotherapeutic PTX is a routine therapy approach; however, developing resistance to PTX has been a major obstacle for chemotherapy failure [24]. In cancer therapy, the effects of this problem frequently result in the recurrence of the disease state [25]. Therefore, elucidating the molecular effects and outcomes of the resistance mechanism is required for identifying potential therapeutic targets to reduce PTX resistance in breast cancer. 

In this study, the PTX-res MCF-7 breast cancer cell line has been generated through increasing PTX concentration treatment over a 6-month period. During the establishment of the PTX-res cell line process, PTX-res MCF-7 cells gained a remarkable different morphology compared to wt MCF-7 cells. In line with this, both untreated and PTX-treated PTX-res MCF-7 cells showed a significant MDR1/ABCB1 protein expression profile compared to wt cells [26] (Figure 1B). Similar to our findings, PTX-res MCF-7 cells showed increased MDR1 activation and protein expression levels linked with the EMT signaling pathway [27]. The wt MCF-7 cells showed an epithelial cell profile with their apicobasal polarity formation and close contact with each other. In contrast, PTX-res MCF-7 cells showed mesenchymal properties, loosely arranged in a three-dimensional extracellular matrix (ECM), and a mesenchymal star-like shape [28,29]. 

The 300-nM PTX-treated MCF-7 PTX-resistant and 100-nM PTX-treated SK-BR-3 PTX-resistant breast cancers showed enhanced survival potential and proliferation ratio in a time-dependent manner [30]. In our results, wt MCF-7 cells were more sensitive to PTX treatment, and wt cells showed a decreased proliferation ratio compared with PTX-res MCF-7 cells in time-dependent PTX treatment (Figure 2A). It was observed that PTX has a suppressive role in the colony formation potential of several breast cancer cells such as MCF-7, BT474, MDA-MB-468, and MDA-MB-231 [31]. In our study, wt MCF-7 cells showed a colony formation decrease similar to previous findings, but PTX-res MCF-7 cells did not show colony formation reduction in PTX treatment (Figure 2B). Similarly, it has been observed that the formation of colonies in 3D environments showed similar results. We suggest that PTX-res MCF-7 cells are more capable of anchorage-independent growth in time-dependent PTX-treatment compared to wt MCF-7 cells (Figure 2C). The enhanced survival profiles of both untreated and PTX-treated MCF-7 cells were visualized, with a decreased cell death number, detected with PI, and increased mitochondrial membrane potential, detected with DiOC6 intensity, compared to wt MCF-7 cells. 

The active PI3K/Akt pathway is associated with the enhanced resistant phenotype of cancer cells [32,33]; our results showed that untreated and treated PTX-res MCF-7 cells upregulated PI3K expression (Figure 3). Phosphorylation of Akt at Thr308 by PDK1 and at Ser473 by mTOR complex 2 (mTORC2) is required for the full activation of this protein [34]. We observed from our results that Akt was fully active in untreated wt and both untreated and PTX-treated PTX-res MCF-7 cells as shown with the significantly upregulated phospho-PDK1 and phospho-Akt Ser473 levels. The phosphorylation of GSK3β at Ser9 can be regulated by PI3K/Akt, MAPK, and Wnt signaling pathways. The inhibitory phosphorylation of GSK3β at Ser9 is associated with cell differentiation and chemoresistance [35]. The decreased phosphorylation of GSK3β at Ser9 was observed in PTX-res MCF-7 cells compared to wt cells which confirmed the resistant potential of generated PTX-res cell models. As a downstream target of active Akt, the expression of Cyclin D1 decreased in PTX-treated wt MCF-7 cells in relation to taxol reduced Cyclin D1 expression in ovarian tumors [36]. However, Cyclin D1 expression, which is associated with the cell cycle, still showed expression in both untreated and treated PTX-res MCF-7 cells, which confirmed the enhanced survival potential of these cells. 

The studies about the role of MAPK signaling in breast cancer resistance showed that p38 and p44/42 proteins are activated by stress or growth factor stimuli in drug-resistant conditions [37,38]. Our data indicate that MAPK signaling was initiated with the active Raf signaling and continued with mitogen-activated p38 and p44/42 activation in both untreated and treated PTX-res MCF-7 cells compared to wt cells, although PTX slightly decreased their expression profiles. Stress activated protein kinase SAPK/JNK was significantly active in untreated PTX-res MCF-7 cells compared to wt cells. We suggest that the increased survival potential of PTX-resistant cell lines was caused by the stress-mediated activation of Akt due to the remarkable increase in p38 and Akt activation, which was clearly observed in our results [39]. We report that continuous cell division, as observed with the presence of Cyclin D expression, as a cell cycle initiator both in untreated and treated PTX-res cells, improved the growth and progression of PTX-res MCF-7 cells [40]. According to the literature, the downregulation and/or silencing of Cyclin D1 enhances the migratory properties of MDA-MB-231 and MDA-MB-452 cells in an ERK1/2- and CDK4/6-independent manner [41].

The acquired PTX resistance due to the induction of EMT has a significant role in the formation of resistant malignant cells [42]. Consistent with the previous results in the triggering of EMT conditions in PTX resistance, both untreated and PTX-treated PTX-res MCF-7 cells showed diminished epithelial E-cadherin and Claudin expression, however, the increased expression of TCF8/ZEB1 was a transcription factor [27]. The upregulation of TCF8/ZEB1 was associated with increased mesenchymal markers such as N-cadherin, Vimentin, and Snail in Figure 4A [43,44]. In normal epithelial cells, E-cadherin is associated with the cytoskeleton through the binding of β-catenin. The loss of E-cadherin causes the release and translocation of β-catenin into the nucleus to induce the transcription of mesenchymal marker genes such as TWIST [45]. Although the expression level of β-catenin did not show a significant change in our results, the increase in Twist mRNA expression, which was shown in supplemental Figure 2, confirmed the loss of E-cadherin-mediated β-catenin-induction. The diminished expression of the tumor suppressor and Wnt signaling inhibitor Axin-1 protein in both untreated and PTX-treated PTX-res MCF-7 cells confirmed its inhibitive role on proliferation, and EMT in cancer [46]. The triggering of EMT in the resistance mechanism may be related to the active oncogenic Akt pathway, which directly represses E-cadherin and produces transcriptional factor Snail, a repressor of E-cadherin [47]. The mesenchymal and invasive potential of PTX-res MCF-7 cells was confirmed by the wound-closure capability in a time-dependent manner (Figure 4B). 

JNK-IN-8 is one of the first irreversible JNK inhibitors to form a covalent bond with a conserved cysteine and has been described as an extremely potent enzymatic and cellular JNK inhibitor that directly inhibits c-Jun, the phosphorylation substrate [48,49]. Following significant SAPK/JNK induction in our PTX-res MCF-7 cell, we used JNK inhibitor to evaluate the functional role of JNK inhibition on the sensitization of PTX-res MCF-7 cells. According to the literature, PTX treatment activates c-Jun N-terminal kinase (JNK), which was reversed in the presence of JNK-IN-8 [50]. Phospho-p38 expression and SAPK/JNK expression remarkably decreased upon JNK inhibition in untreated and treated wt and PTX-res MCF-7 cells due to destabilization of the protein [51]. In our results, JNK inhibition may lead to destabilization of the cascade and prevent the activation of p38 [52,53]. Although E-cadherin expression was not detected in PTX-res MCF-7 cells, the mesenchymal marker Vimentin expressions was abolished in the JNK inhibitor alone or in the combined JNK inhibitor-treated wt and PTX-res MCF-7 cells (Figure 5B). However, cells were still able to express MDR/ABCB1 protein; therefore, we suggest that the PTX resistance was affected due to other signaling pathways in addition to EMT. 

It is known that the Wnt/β-catenin pathway is involved in cancer progression, development, and EMT [54]. During the loss of Wnt signals, β-catenin is found within a complex of adenomatous polyposis coli (APC) and Axin, following the phosphorylation of β-catenin by GSK3β the ubiquitin-mediated degradation of β-catenin occurs. However, in the active Wnt/β-catenin signaling, Wnt molecules bind to Frizzled receptors and low-density, lipoprotein-receptor-related protein 5/6 (LRP5/6) which inactivated GSK3β and inhibits the degradation of β-catenin and allows β-catenin to accumulate and translocate to the nucleus for the transcriptional regulation of EMT genes through acting as a co-activator for T-cell factor (TCF)/lymphoid enhancer factor (LEF) [32,55]. The target genes of the translocation of β-catenin into nucleus-mediated active canonical or non-canonical Wnt signaling, such as c-JUN, c-MYC, Cyclin D1, MDR1/ABCB1, MMPs, Axin-1, 2, have a role in the regulation of cell survival, proliferation, and differentiation [56,57]. However, Wnt signaling pathway inhibitors have gained importance for reversing drug-resistance in cancer cells [58]. 

The expressions of the canonical Wnt signaling pathway member Wnt3a and non-canonical pathway member Wnt5a were upregulated in untreated and treated-PTX-res cells compared to wt MCF-7 cells. Our result confirmed the studies that Wnt3a triggered β-catenin-mediated EMT and Wnt5a/b-mediated cell polarity and motility [59]. Treatment with JNK inhibitors decreased the expression profiles of both Wnt3a and Wnt5a/b in untreated or PTX-treated wt and PTX-res MCF-7 cells, while Wnt5a showed a more downregulated expression profile compared to Wnt3a expression. This might be associated with the previous results that showed that Wnt5a had a role in the activation of JNK to induce Wnt5a-JNK-mediated cell polarization and migration [60]. The significant phosphorylation of the co-receptor of the Wnt signaling pathway, LDL-receptor-related protein 6 (LRP6), at Ser1490 in untreated and PTX-treated PTX-res MCF-7, confirmed the activation of Wnt signaling cascade. However, JNK inhibition remarkably decreased the phosphorylation of LRP6 at Ser1490 which might result in the suppression of β-catenin- mediated EMT signaling. A similar result was observed in total LRP6. Dvl proteins, which are homologous of Dishevelled (Dsh) proteins, are important mediators for the canonical and non-canonical Wnt signaling pathway [55]. Active Wnt signaling stimulation induces Dvl protein activation and the activation of this scaffold protein triggers the β-catenin-TCF/LEF interaction in the nucleus to trigger Wnt signaling pathway genes [61,62]. Untreated and PTX-treated wt MCF-7 cells showed diminished expression of Dvl2 and Dvl3, JNK inhibition did not show significant change. Similar to the other Wnt signaling members, Dvl2 and Dvl3 expression were significantly upregulated in both untreated and PTX-treated PTX-res MCF-7 cells, however, JNK inhibition has a remarkable suppressive effect on the expression profiles of these proteins. Previous studies showed that Axin has a dual role in Wnt signaling, which has a role in the β-catenin destruction complex to degrade β-catenin and the inhibition of Wnt; it also promotes LRP5/6 phosphorylation to induce Wnt signaling activation [63]. Wnt-related regulation of survival in PTX resistant colonies was also shown in MDA-MB-231 breast cancer cells by GSE12791 [64]. According to GSE12791, while FOS expression levels were increasing in PTX-resistant MDA-MB-231 cells, Wnt1 inhibitor DKK and Cadherin were downregulated. Our results showed Axin-1 upregulation in untreated and PTX-treated PTX-res MCF-7 cells compared to wt MCF-7 cells, but the co-treatment of JNK inhibitor with PTX downregulated its expression in PTX-res MCF-7 cells. Therefore, we suggested that Axin-1 showed a Wnt signaling activator role associated with the elevated LRP6 phosphorylation in PTX-res MCF-7 cells, and the expression of Axin1 was downregulated by co-treatment with JNK and PTX according to the drug-resistant reverse effect of the JNK inhibitor. 

## 5. Conclusions

We conclude that our results confirm that cellular motility profile was higher in PTX-res MCF-7 breast cancer cells compared to wt cells. Therefore, PTX-resistant cells showed more aggressive characteristics than PTX-sensitive breast cancer cells. This situation might negatively affect the patient’s survival and treatment success in therapy. The application of JNK-IN-8 indicated that, although the drug resistance marker was expressed in PTX-res cells, the decrease in the Wnt signal pathway might inhibit migration and motility in PTX-res MCF-7 cells. In future studies, by using potential inhibitors such as JNK-IN-8, cancer progression might be reduced even in a difficult mechanism such as drug resistance as a promising cancer therapy approach.

## Figures and Tables

**Figure 1 biology-09-00320-f001:**
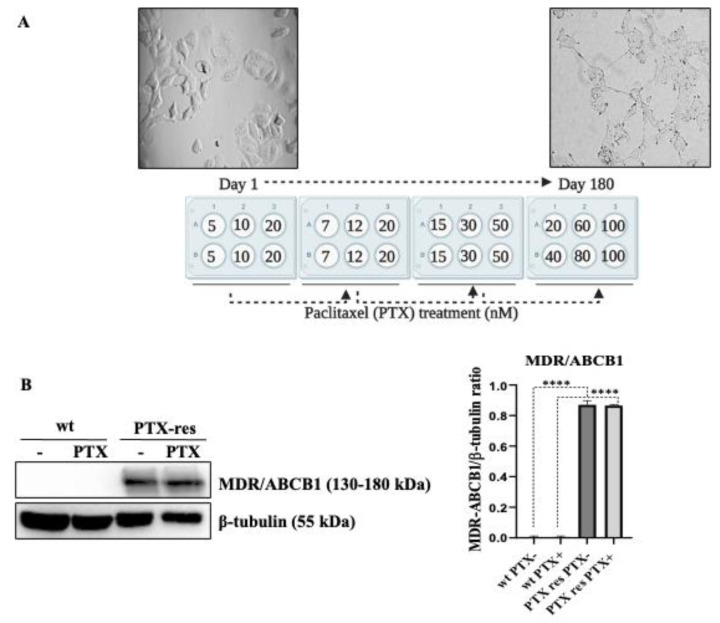
Generation and determination of Paclitaxel (PTX) resistance in MCF-7 breast cancer cells. (**A**) Schematic representation of the development of PTX-resistant cells in MCF-7 cells. MCF-7 cells were treated with 5–100 nM over a time period of 6 months with an increasing concentration of PTX treatment. Morphological changes between wt and PTX-res MCF-7 cells against 100 nM. (**B**) The expression profile of multidrug resistance (MDR)/ABCB1 protein was determined by immunoblotting. Densitometric analysis was shown on the right panel. β-tubulin was selected as a loading control. **** *p* < 0.0001.

**Figure 2 biology-09-00320-f002:**
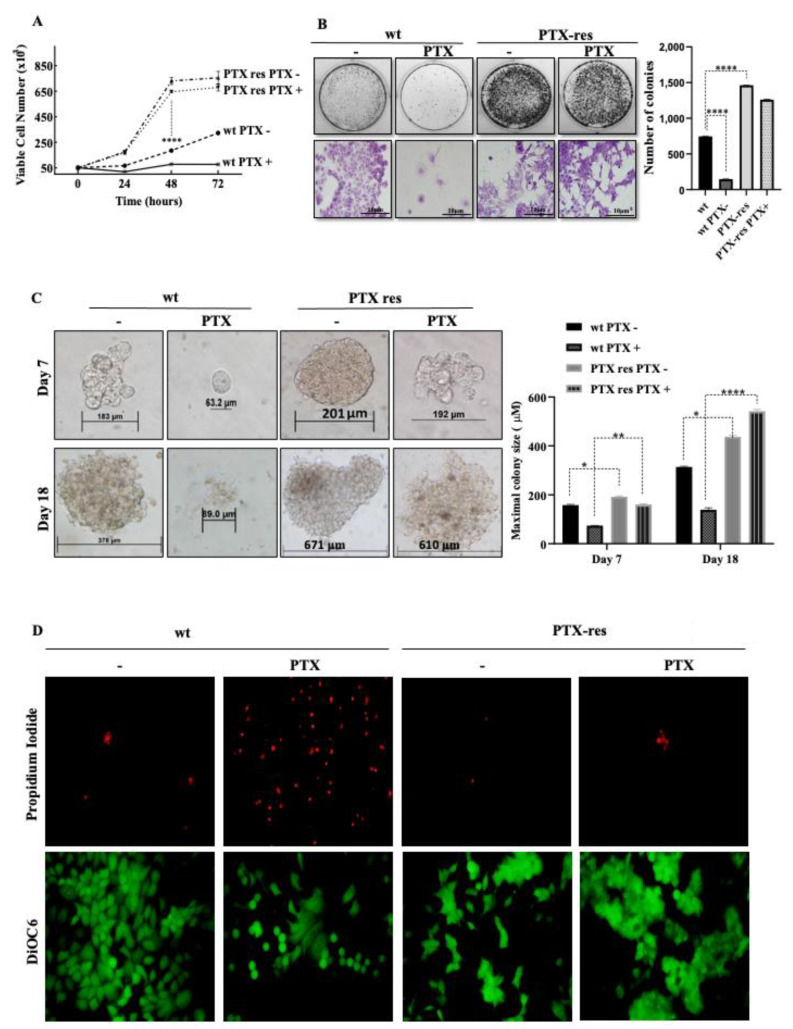
The effect of PTX resistance on MCF-7 cell sensitivity. (**A**) PTX-resistance enhanced the proliferation ratio of MCF-7 cells determined by the Trypan Blue Dye Exclusion assay. (**B**) The colony formation potentials of PTX-res MCF-7 were significantly higher compared to wt cells determined by colony formation assay. Purple colonies determined the magnified view of specific colonies of the upper panel. (**C**) PTX-res MCF-7 cells enclosed in soft agar showed the progressive formation of colonies rather than wt MCF-7 parental cells at the end of day 18. Columns presented the mean of colony sizes of three independent experiments within the 7th-14th-18th days. (**D**) PI fluorescence dye staining related to cellular death and mitochondrial membrane potential following DiOC6 fluorescent staining in PTX-res cells treated/untreated paclitaxel in wt. Differences between the death and survival of wt and PTX-res cells were observed due to paclitaxel treatment. * *p* < 0.05; ** *p* < 0.001; **** *p* < 0.0001.

**Figure 3 biology-09-00320-f003:**
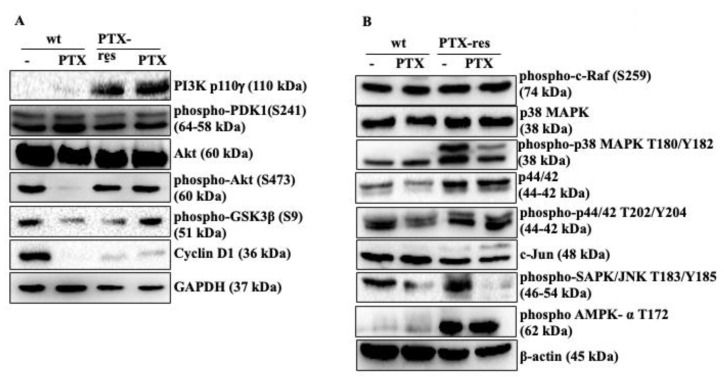
The effect of PTX resistance on PI3K/Akt and MAPK pathways. (**A**) The expression profiles of PI3K/AKT and (**B**) MAPK signaling pathway members were determined in untreated and PTX-treated PTX-res MCF-7 cells by immunoblotting assay. β-actin and GAPDH were used as loading control. Densitometry analysis was shown at Appendix A from different batches of PTX-resistant MCF-7 cells.

**Figure 4 biology-09-00320-f004:**
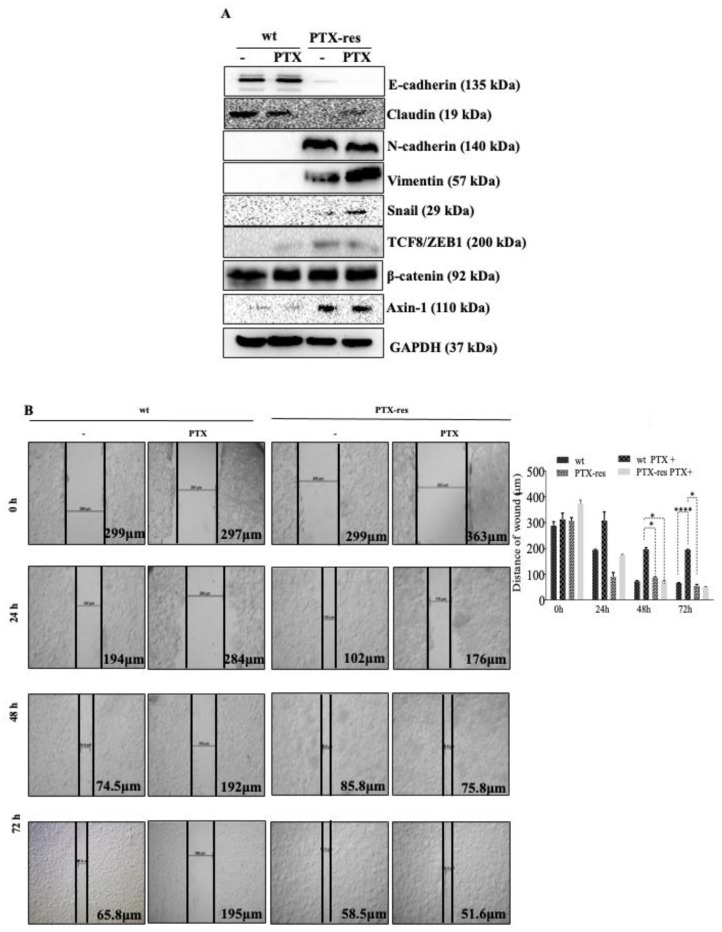
Examination of motility profile of PTX-res MCF-7 cells related to EMT pathway. (**A**) The effect of PTX resistance on the EMT pathway determined by immunoblotting. GAPDH was used as a loading control. Densitometry analysis was shown at Appendix A from different batches of PTX-resistant MCF-7 cells. (**B**) PTX-res cells showed higher wound closure capacity than wt parental MCF-7 cells in a time-dependent manner investigated by wound-healing assay. * *p* < 0.05; **** *p* < 0.0001.

**Figure 5 biology-09-00320-f005:**
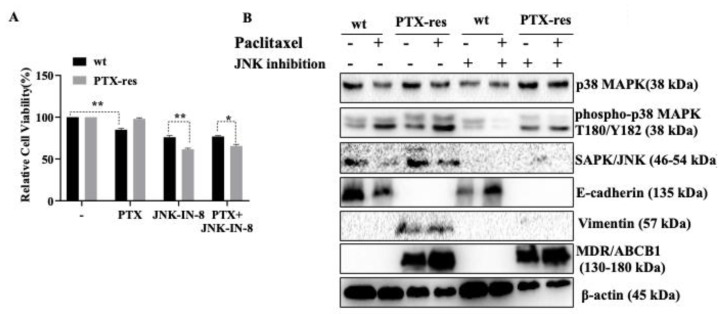
Effects of individual and combined treatment of JNK-IN-8 on cell viability and cellular survival of wt and PTX-res MCF-7 cells. (**A**) The cell viability of wt and PTX-res MCF-7 cells was determined by MTT cell viability assay following treatment of individual PTX or JNK inhibitor and combination of both agents. (**B**) The expression profiles of p38 MAPK, p-p38 MAPK, SAPK/JNK, p-SAPK/JNK, E-cadherin, Vimentin, and MDR/ABCB1 gene were determined by immunoblotting in wt and PTX-res MCF-7 breast cancer cells following 24 h PTX and JNK-IN-8 treatment. β-actin was used as a loading control. Densitometry analysis was shown at Appendix A from different batches of PTX-resistant MCF-7 cells. * *p* < 0.05; ** *p* < 0.001.

**Figure 6 biology-09-00320-f006:**
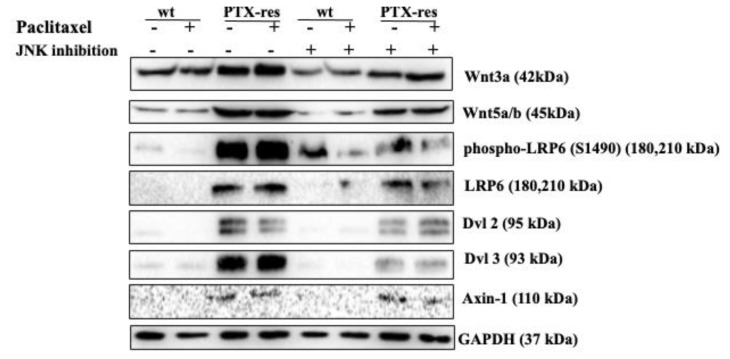
Investigation of the effect of JNK-IN-8 on invasion and metastasis in wt and PTX-res MCF-7 cells associated with the Wnt signaling pathway. The expression profiles of Wnt3a, Wnt5a/b, phospho-LRP6, LRP6, Dvl2, Dvl3, Axin-1 were determined by immunoblotting in wt and PTX-res MCF-7 breast cancer cells. GAPDH was used as a loading control. Densitometry analysis was shown at Appendix A from different batches of PTX-resistant MCF-7 cells.

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
