# Peer review of "Specific c-Jun N-Terminal Kinase Inhibitor, JNK-IN-8 Suppresses Mesenchymal Profile of PTX-Resistant MCF-7 Cells through Modulating PI3K/Akt, MAPK and Wnt Signaling Pathways"

_biology, 2020, doi:10.3390/biology9100320_

Round 1
Reviewer 1 Report
The work by Kilbas et al., reports that an inhibitor of the JNK affects the profile of the taxol derived drug placlitaxel (PTX) resistant cell line which the authors stablish in this work. The model that they use is the the breast cancer line MCF-7. The PTX resistant cells have a higher cell survival, enhanced colony formation and exhibit a higher cell migration. These cells have properties found in cells that have suffer EMT as well as an over activation of the PI3K/Akt and MAPK pathways. Based in these observations the authors analyze the effect of a JNK inhibitor, JNK-In-8 on the cells, observing a suppression in these pathways. Furthermore, they also show a downregulation in the Wnt pathway that is over activated in the PTX-resistance cells. With these results the authors propose that this JNK inhibitor may be a good candidate to treat cancer cells that are resistant to PTX.
In general, the results are clear, although some figures require higher quality. However, this work lacks originality since there are many published works that show similar results and conclusions, as the ones reported in this work. In some of these studies, also an MCF-7 resistance cell line to PTX is used as model. Many of these publications are not citated in this manuscript or discussed in this report. An example of these publications is included in this revision. Therefore and based in these observations I do not recommend the acceptance of this work in Biology.
Minor points:
1.- In the beginning of the results section, page 6, line 218 not a subtitle is indicated.
2.- In figure 2D, it is important to show a merge of the propidium iodide and DiOC6 staining’s panels, since its colocalization it is not evident.
3.- In figure 4 the differences in wound closure are difficult to see. The image requires better resolution.
4.- Along the manuscript some references are cited with different formats. Also, some references are not cited.
Some related publications:
Paclitaxel resistance in MCF-7/PTX cells is reversed by paeonol through suppression of the SET/phosphatidylinositol 3-kinase/Akt pathway.
Zhang W, Cai J, Chen S, Zheng X, Hu S, Dong W, Lu J, Xing J, Dong Y .Mol Med gularion 11.PMID: 25760096
In addition, paeonol inhibited SET-mediated paclitaxel resistance by attenuating PI3K/Akt pathway activity in the MCF-7/PTX cells. In conclusion, the results of the present study demonstrated that SET was associated with paclitaxel resistance
SET protein overexpression contributes to paclitaxel resistance in MCF-7/S cells through PI3K/Akt pathway.
Zhang W, Zheng X, Meng T, You H, Dong Y, Xing J, Chen S.J Drug Target. 2017 Mar;25(3):255-263. doi: 10.1080/1061186X.2016.1245307. Epub 2016 Oct 20.PMID: 27718638
A microarray based expression profiling of paclitaxel and vincristine resistant MCF-7 cells.
Kars MD, IÅŸeri OD, Gündüz U.Eur J Pharmacol. 2011 Apr 25;657(1-3):4-9. doi: 10.1016/j.ejphar.2011.02.001. Epub 2011 Feb 18.PMID: 21320484
Inhibitors of PI3K/ERK1/2/p38 MAPK Show Preferential Activity Against Endocrine-Resistant Breast Cancer Cells.
Khaj Mathew PM, Luqmani YA.
Drug resistant MCF-7 cells exhibit epithelial-mesenchymal transition gene expression pattern.
IÅŸeri OD, Kars MD, Arpaci F, Atalay C, Pak I, Gündüz U.Biomed Pharmacother. 2011 Feb;65(1):40-5. doi: 10.1016/j.biopha.2010.10.004. Epub 2010 Nov 4.PMID: 21177063
MiR-125b regulates epithelial-mesenchymal transition via targeting Sema4C in paclitaxel-resistant breast cancer cells.
Yang Q, Wang Y, Lu X, Zhao Z, Zhu L, Chen S, Wu Q, Chen C, Wang Z.Oncotarget. 2015 Feb 20;6(5):3268-79.
Acquisition of epithelial-mesenchymal transition is associated with Skp2 expression in paclitaxel-resistant breast cancer cells.
Yang Q, Huang J, Wu Q, Cai Y, Zhu L, Lu X, Chen S, Chen C, Wang Z.Br J Cancer. 2014 Apr 15;110(8):1958-67. doi: 10.1038/bjc.2014.136. Epub 2014 Mar 18.
Down-regulation of Raf-1 kinase is associated with paclitaxel resistance in human breast cancer MCF-7/Adr cells.
Lee M, Koh WS, Han SS.Cancer Lett. 2003 Apr 10;193(1):57-64. Do
HIF-2α promotes conversion to a stem cell phenotype and induces chemoresistance in breast cancer cells by activating Wnt and Notch pathways.
Yan Y, Liu F, Han L, Zhao L, Chen J, Olopade OI, He M, Wei M.J Exp Clin Cancer Res. 2018 Oct 19;37(1):256. doi: 10.1186/s13046-018-0925-x.
The transcriptional coactivator WBP2 primes triple-negative breast cancer cells for responses to Wnt signaling via the JNK/Jun kinase pathway.
Li Z, Lim SK, Liang X, Lim YP.J Biol Chem. 2018 Dec 28;293(52):20014-20028. doi: 10.1074/jbc.RA118.005796. Epub 2018 Nov 15.
Hypoxia counteracts taxol-induced apoptosis in MDA-MB-231 breast cancer cells: role of autophagy and JNK activation.
Notte A, Ninane N, Arnould T, Michiels C.Cell Death Dis. 2013 May 16;4(5):e638.
Inhibition of JNK reduces G2/M transit independent of p53, leading to endoreduplication, decreased proliferation, and apoptosis in breast cancer cells.
Mingo-Sion AM, Marietta PM, Koller E, Wolf DM, Van Den Berg CL.Oncogene. 2004 Jan 15;23(2):596-604.
Down-regulation of Raf-1 kinase is associated with paclitaxel resistance in human breast cancer MCF-7/Adr cells.
Lee M, Koh WS, Han SS.Cancer Lett. 2003 Apr 10;193(1):57-64.
Author Response
We are thankful for the valuable comments.
Minor points:
1.- In the beginning of the results section, page 6, line 218 not a subtitle is indicated.
The subtitle of page 6 line 218 (in the old version) is indicated at line 220 (updated version) and all the subtitles of the results section is revised.
2.- In figure 2D, it is important to show a merge of the propidium iodide and DiOC6 staining’s panels, since its colocalization it is not evident.
We are unable to show the co localization of PI and DiOC6 staining. However, the intensity of images described the results of the experiment through increased cell death and diminished cell viability due to mitochondrial dysfunction.
3.- In figure 4 the differences in wound closure are difficult to see. The image requires better resolution.
The images are in high resolution but the image quality has decreased because of fitting them on a single slide. Therefore, we prepared a file that contains the original images in high resolution.
4.- Along the manuscript some references are cited with different formats. Also, some references are not cited.
Some related publications:
Paclitaxel resistance in MCF-7/PTX cells is reversed by paeonol through suppression of the SET/phosphatidylinositol 3-kinase/Akt pathway. The reference is cited as number 32 in line 432.
Zhang W, Cai J, Chen S, Zheng X, Hu S, Dong W, Lu J, Xing J, Dong Y .Mol Med gularion 11.PMID: 25760096 The reference
In addition, paeonol inhibited SET-mediated paclitaxel resistance by attenuating PI3K/Akt pathway activity in the MCF-7/PTX cells. In conclusion, the results of the present study demonstrated that SET was associated with paclitaxel resistance The reference to this study is cited in the same format.
SET protein overexpression contributes to paclitaxel resistance in MCF-7/S cells through PI3K/Akt pathway.
Zhang W, Zheng X, Meng T, You H, Dong Y, Xing J, Chen S.J Drug Target. 2017 Mar;25(3):255-263. doi: 10.1080/1061186X.2016.1245307. Epub 2016 Oct 20.PMID: 27718638 The reference is cited as number 33 in the line 448.
A microarray based expression profiling of paclitaxel and vincristine resistant MCF-7 cells.
Kars MD, IÅŸeri OD, Gündüz U.Eur J Pharmacol. 2011 Apr 25;657(1-3):4-9. doi: 10.1016/j.ejphar.2011.02.001. Epub 2011 Feb 18.PMID: 21320484 The reference is cited as number 26 in line 427.
Inhibitors of PI3K/ERK1/2/p38 MAPK Show Preferential Activity Against Endocrine-Resistant Breast Cancer Cells.
Khaj Mathew PM, Luqmani YA.- The reference is cited as number 51 in line 500.
Drug resistant MCF-7 cells exhibit epithelial-mesenchymal transition gene expression pattern.
IÅŸeri OD, Kars MD, Arpaci F, Atalay C, Pak I, Gündüz U.Biomed Pharmacother. 2011 Feb;65(1):40-5. doi: 10.1016/j.biopha.2010.10.004. Epub 2010 Nov 4.PMID: 21177063- The reference is cited as number 43 in line 480.
MiR-125b regulates epithelial-mesenchymal transition via targeting Sema4C in paclitaxel-resistant breast cancer cells.
Yang Q, Wang Y, Lu X, Zhao Z, Zhu L, Chen S, Wu Q, Chen C, Wang Z.Oncotarget. 2015 Feb 20;6(5):3268-79. The reference is cited as number 4 in the Introduction part line 39.
Acquisition of epithelial-mesenchymal transition is associated with Skp2 expression in paclitaxel-resistant breast cancer cells.
Yang Q, Huang J, Wu Q, Cai Y, Zhu L, Lu X, Chen S, Chen C, Wang Z.Br J Cancer. 2014 Apr 15;110(8):1958-67. doi: 10.1038/bjc.2014.136. Epub 2014 Mar 18. - The reference is cited as number 28 in line 432.
Down-regulation of Raf-1 kinase is associated with paclitaxel resistance in human breast cancer MCF-7/Adr cells.
Lee M, Koh WS, Han SS.Cancer Lett. 2003 Apr 10;193(1):57-64. The reference is cited as number 37 in the line 463.
HIF-2α promotes conversion to a stem cell phenotype and induces chemoresistance in breast cancer cells by activating Wnt and Notch pathways.
Yan Y, Liu F, Han L, Zhao L, Chen J, Olopade OI, He M, Wei M.J Exp Clin Cancer Res. 2018 Oct 19;37(1):256. doi: 10.1186/s13046-018-0925-x.- The reference is cited as number 19 in line 80
Notte A, Ninane N, Arnould T, Michiels C.Cell Death Dis. 2013 May 16;4(5):e638. The reference is cited as number 50 in the line 498.
Mingo-Sion AM, Marietta PM, Koller E, Wolf DM, Van Den Berg CL.Oncogene. 2004 Jan 15;23(2):596-604. The reference is cited as number 48 in the line 495.
Down-regulation of Raf-1 kinase is associated with paclitaxel resistance in human breast cancer MCF-7/Adr cells. Lee M, Koh WS, Han SS.Cancer Lett. 2003 Apr 10;193(1):57-64. The reference is cited as number 37 in the line 463

Reviewer 2 Report
This is an important and relevant study
Overall the experiments are well thought out and carried out with correct controls.
There is a significant amount of data which mostly is well presented.
Overall the whole manuscript needs a lot of editing as the manuscirpt often switches from different tenses
Specifically:
Abstract needs rewording. "Is a common cause for breast cancer therapy failure"
"Generated ptx resistant...change to The generated ptx resistant
Intro needs editing- sentence starting with 2018
line 24 - are shown
line 126 superscript not used and throughout document
line 131 Obtained data- reword sentence
line 169 Transferred to
line 227 Did not observe- reword
line 244 Was significantly higher
Fig 2d needs a scale bar
Fig 3 very crowded text on left side of blot
on my copy fig 4 appears disjointed
line 391 sentence does not make sense reword
Very long sentence beginning line 405 needs rewording
Author Response
We are all thankful for the comments.
This is an important and relevant study
Overall the experiments are well thought out and carried out with correct controls.
There is a significant amount of data which mostly is well presented.
Overall the whole manuscript needs a lot of editing as the manuscript often switches from different tenses
Specifically:
Abstract needs rewording. "Is a common cause for breast cancer therapy failure"
Sentence is revised and indicated in line 16.
"Generated ptx resistant...change to The generated ptx resistant
Revised in line 20.
Intro needs editing- sentence starting with 2018-Revised “In 2018” in line 33.
line 24 - are shown-revised in line 24.
line 126 superscript not used and throughout document-revised in line 133 and throughout the manuscript.
line 131 Obtained data- reword sentence- Revised. In line 136.
line 169 Transferred to- revised in line 177.
line 227 Did not observe- reword- revised in line 242.
line 244 Was significantly higher- revised in line 263.
Fig 2d needs a scale bar-The scale bar is added to Figure 2d.
Fig 3 very crowded text on left side of blot-
We are unable to delete the names and kDa values of each blot.
on my copy fig 4 appears disjointed
Figure 4 is shown disjoined since it is located on 2 separate pages.
line 391 sentence does not make sense reword
Line 391 is revised.
Very long sentence beginning line 405 needs rewording-Revised was added as “The downstream members of Wnt signaling pathway, phospho- LRP6 at Ser1490, Dvl2 and Dvl3 and Axin-1 showed similar expression profiles as they upregulated in both PTX untreated/treated PTX-res cells compared to wt cells, however only and combined treatment of JNK inhibitor significantly decrease this upregulation (Figure 6) and indicated in line 436.
Reviewer 3 Report
The authors have explored the mechanisms behind PTX resistance in MCF7 cells. This is an important question, but not entirely novel. Nonetheless, the manuscript is well-structured, and the experiments technically sound.
However, there are several issues that need to be adjusted before being reconsidered for publication.
My major concern is about how the results support the conclusions. For instance, the authors stated "PTX-resistance enhance the invasion and metastatic properties of MCF7 cells". However, the authors demonstrated in any experiment neither the invasiveness nor the metastatic potential of those cells. They simply performed migration and EMT assays which not necessarily means invasion or metastasis, but only migration potential of PTX resistant cells. Therefore, the current version of paragraph 3.3's title is misleading. The same is true for the title of the legend for Figure 4. If the authors want to claim an effect on invasiveness, they need to perform invasion assays with ECM-coated trans-well plates. Metastasis assays are even more complex because they need to involve animal models.
Is this phenotype only visible in MCF7 cells? The authors have proven PTX resistance mechanisms only in a cellular model for breast cancer; to generalize the effect, they should confirm some of the observations also in another similar cellular model (i.e., based on T47D, ZR-75-1, or another cellular system).
Figure 2A: Cyclin D1 levels changed remarkably between wt and resistant cells, and PTX alone strongly reduced it. The authors need to discuss that more thoroughly to speculate about the obtained results.
Figure 2B: The reduction in phospho-SAPK/JNK seems to be unrelated to PTX resistance given that it is down-regulated by PTX also in resistant cells.
JNK inhibitor alone almost completely reduced the levels of phospho-p38 MAPK. How do you explain this? Resistant cells also recover from this inhibition. Any speculation on how can this happen?
JNK inhibitor determined higher levels of phospho-LRP6; this is something that needs to be clearly discussed in the text.
Lastly, given that JNK inhibitor is known to block its enzymatic activity on the targets (i.e., c-Jun), why was this not tested? The authors showed a strong reduction in the total levels of SAPK/JNK. Is this something expected? The authors need to comment on that.
Minor issues:
- There are several typos that need to be adjusted (i.e., several superscripts were wrongly written, missing spaces, etc etc). Moreover, some sentences are convoluted. Please, revise the manuscript carefully.
- Line 127: what does it mean I quote the authors' words "treated with 100nM PTX at 24h within 72hr"??
- Page 5: despite the concentration, it would be more useful to include the clone of the used antibodies.
- Line 218: That sentence seems a title but written as it is, it doesn't look like this.
- Missing verbs in the sentences at lines 243-244; 255-256; 281-282; 298-299; 362-363.
- Line 258: change "red-stained cells" with "red-fluorescent cells", given that PI is a fluorescent dye.
- Figure 2: Some captions are too small and difficult to read; there is no need to add twice "Figure 2" on the right of panels B and D (same for other Figures).
- Panel 2B: If the lower panel is a magnification (as I imagine) of the upper panel, it has to be indicated into the legend.
- Line 300, about Figure 3: how can the data be statistically significant if the numerosity is = 1?
- Some problems with the references. At line 479 a reference is missing. The same at line 505. Line 511-512 a reference is cited in an unformatted version.
Author Response
We are all thankful for the valuable comments. We try to address each issue one by one.
The authors have explored the mechanisms behind PTX resistance in MCF7 cells. This is an important question, but not entirely novel. Nonetheless, the manuscript is well-structured, and the experiments technically sound.
However, there are several issues that need to be adjusted before being reconsidered for publication.
My major concern is about how the results supports the conclusions. For instance, the authors stated "PTX-resistance enhance the invasion and metastatic properties of MCF7 cells". However, the authors demonstrated in any experiment neither the invasiveness nor the metastatic potential of those cells. They simply performed migration and EMT assays which not necessarily means invasion or metastasis, but only migration potential of PTX resistant cells. Therefore, the current version of paragraph 3.3's title is misleading.
The title of 3.4 is revised in line 313.
The same is true for the title of the legend for Figure 4. If the authors want to claim an effect on invasiveness, they need to perform invasion assays with ECM-coated trans-well plates. Metastasis assays are even more complex because they need to involve animal models.
The title of the legend for Figure 4 is revised.
Is this phenotype only visible in MCF7 cells? The authors have proven PTX resistance mechanisms only in a cellular model for breast cancer; to generalize the effect, they should confirm some of the observations also in another similar cellular model (i.e., based on T47D, ZR-75-1, or another cellular system).
We are planning to confirm our studies in different cellular models in future studies.
Figure 2A: Cyclin D1 levels changed remarkably between wt and resistant cells, and PTX alone strongly reduced it. The authors need to discuss that more thoroughly to speculate about the obtained results.
More evidence on the relationship between the silencing of Cyclin D and cell migratory profile of breast cancer cells was added to the Discussion section as “According to the literature the downregulation or silencing of Cyclin D1 enhanced the migration properties of MDA-MB-231 and MDA-MB-452 cells in ERK1/2 and CDK4/6 independent manner. “ This study is cited as number 41 in ine 474.
Figure 2B: The reduction in phospho-SAPK/JNK seems to be unrelated to PTX resistance given that it is down-regulated by PTX also in resistant cells.-
The selected JNK inhibitor treatment may lead to destabilization of the signaling cascade and prevent the activation of p38 as shown in previous studies.
Sunters, Andrew & Madureira, Patrícia & Pomeranz, Karen & Aubert, Muriel & Brosens, Jan & Cook, Simon & Burgering, Boudewijn & Coombes, R & Lam, Eric. (2006). Paclitaxel-Induced Nuclear Translocation of FOXO3a in Breast Cancer Cells Is Mediated by c-Jun NH2-Terminal Kinase and Akt. Cancer research. 66. 212-20. 10.1158/0008-5472.CAN-05-1997.
Notoamide-type alkaloid induced apoptosis and autophagy via a P38/JNK signaling pathway in hepatocellular carcinoma cells. DOI: 10.1039/C9RA03640G (Paper) RSC Adv., 2019, 9, 19855-19868
These references were added to the Discussion section as numbered 52 and 53 in line 590.
JNK inhibitor determined higher levels of phospho-LRP6; this is something that needs to be clearly discussed in the text.
It is colored by red and revised in line 623.
Lastly, given that JNK inhibitor is known to block its enzymatic activity on the targets (i.e., c-Jun), why was this not tested? The authors showed a strong reduction in the total levels of SAPK/JNK. Is this something expected? The authors need to comment on that.-
The inhibitor and its effect were already known from our previous studies and unpublished data.
- https://www.selleckchem.com/JNK.html
- Arisan ED, Rencuzogullari O, Keskin B, Grant GH, Uysal-Onganer P. Inhibition on JNK Mimics Silencing of Wnt-11 Mediated Cellular Response in Androgen- Independent Prostate Cancer Cells. Biology (Basel). 2020 Jun 27;9(7):E142. doi:10.3390/biology9070142. PMID: 32605008.
- Koushyar S, Grant GH, Uysal-Onganer P. The interaction of Wnt-11 and signalling cascades in prostate cancer. Tumour Biol. 2016;37(10):13049-13057. doi:10.1007/s13277-016-5263-z
Minor issues:
- There are several typos that need to be adjusted (i.e., several superscripts were wrongly written, missing spaces, etc etc). Moreover, some sentences are convoluted. Please, revise the manuscript carefully.
- The manuscript is revised.
- Line 127: what does it mean I quote the authors' words "treated with 100nM PTX at 24h within 72hr"?? “at 24h” is deleted in line 132 (old version 127, new version 132).
- Page 5: despite the concentration, it would be more useful to include the clone of the used antibodies.
The clones of used antibodies were added to the manuscript.
- Line 218: That sentence seems a title but written as it is, it doesn't look like this.
- Line 218 in old version is revised and indicated in line 233.
- Missing verbs in the sentences at lines 243-244; 255-256; 281-282; 298-299; 362-363.
- Missing verbs at specified lines were added to the manuscript.
- Line 258: change "red-stained cells" with "red-fluorescent cells", given that PI is a fluorescent dye.
- Line 258 is revised in line 277..
- Figure 2: Some captions are too small and difficult to read; there is no need to add twice "Figure 2" on the right of panels B and D (same for other Figures).-
- The captions of all figures were deleted.
- Panel 2B: If the lower panel is a magnification (as I imagine) of the upper panel, it has to be indicated into the legend.
“Purple colonies determined the magnified view of a specific colonies of the upper panel.” was added to figure legend.
Line 300, about Figure 3: how can the data be statistically significant if the numerosity is = 1? .
Tne line 300 is determined to be misspelled and corrected to 2.
- Some problems with the references. At line 479 a reference is missing. The same at line 505. Line 511-512 a reference is cited in an unformatted version. - References is added to the manuscript.
Round 2
Reviewer 1 Report
The authors ignored my main concern which is the lack of originality of the work. They do not mention anything to change my opinion that the contributions to the field are minimal and therefore I maintain my position that the article is not suitable for publication.
Author Response
Reply to Reviewer 1
We have already revised the manuscript according to the reviewer’s first-round comments.
At the current phase, following the second round comments, we believe the existing model put forward the importance of the different cellular signaling pathways in drug resistance phenomena. Although the comment was to be not novel, we would like to explain the discussion part in this reply letter:
Although JNK is a well-characterized target in cancer therapy and resistance, there is no direct evidence placed in the literature clearly until now to discuss Wnt-JNK and cell survival/ metastasis potential of the paclitaxel (ptx) resistant cells. Therefore, we believe that the content of the experimental setup includes the novel data on paclitaxel resistance mechanism in breast cancer therapy. The suggested previous literature work did not include a direct scope of our experimental work. However, we revised the paper by addressing the previous studies to our discussion in line with the reviewer’s comments. We appreciate the suggestions that the reviewer provided which we believe gave us more opportunity to discuss the paper in a better way.
Existing literature data points out that Wnt signaling could be critical in the assessment of ptx resistance in different cell lines. According to the previous study, nasopharyngeal carcinoma cells showed marker lncRNA differential expressions which might be critical on Wnt signaling (1).
Ren, S., Li, G., Liu, C., Cai, T., Su, Z., Wei, M., She, L., Tian, Y., Qiu, Y., Zhang, X., Liu, Y., Wang, Y."Next generation deep sequencing identified a novel lncRNA n375709 associated with paclitaxel resistance in nasopharyngeal carcinoma". Oncology Reports 36.4 (2016): 1861-1867.
Additionally, we have revised the discussion part as stated below:
JNK-IN-8 is one of the first irreversible JNK inhibitors to form a covalent bond with a conserved cysteine and has been described as an extremely potent enzymatic and cellular JNK inhibitor that directly inhibits c-Jun, the phosphorylation substrate [48], [49]. Following significant SAPK/JNK induction in our PTX-res MCF-7 cell, we used JNK inhibitor to evaluate the functional role of JNK inhibition on the sensitization of PTX-res MCF-7 cells. According to the literature, PTX treatment activates c-Jun N-terminal kinase (JNK), which was reversed in the presence of JNK-IN-8 [50]. Phospho-p38 expression and SAPK/JNK expression remarkably decreased upon JNK inhibition in untreated and treated wt and PTX-res MCF-7 cells due to the destabilization of the protein [51]. In our results, JNK inhibition may lead to destabilization of the cascade and prevent the activation of p38 [52], [53]. Although E-cadherin expression was not detected in PTX-res MCF-7 cells, the mesenchymal marker Vimentin expressions were abolished in only or when it was combined JNK inhibitor-treated wt and PTX-res MCF-7 cells (Figure 5B). However, cells were still able to express MDR/ABCB1 protein, therefore we suggest that the PTX resistance was affected due to other signaling pathways addition to EMT.
We are again thankful for the suggestions
Best regards
Reviewer 3 Report
The revised version of the manuscript has improved in comparison to the original submitted one.
However, some of the suggestions have been ignored or not adequately addressed by the authors.
Given that the novelty is not very high, my concerns about the lack of confirmatory experiments in at least another cellular model for breast cancer remain. Some crucial experiments need to be also performed in at least another cell line in order to render more general at least some of the observed effects. This is necessary to support acceptance.
I don't mean the authors need to re-generate another cell line system (although it could have been very informative) resistant to Paclitaxel (which can be time-consuming). Still, some observations on PTX-treated cells can be confirmed in other breast cancer-derived cell lines.
Alternatively, some of the experiments might be confirmed in another clone of MCF7 cells resistant to Paclitaxel (as I assume the authors obtained more than one single clone resistant to Paclitaxel).
Moreover, my concerns about "invasiveness" and "metastasis" are still in the text and need to be adjusted throughout the manuscript. As I already mentioned in my previous report, the author did not address anything which supports effects on invasion or metastasis, but, still, this remains cited from the Abstract to Results, Discussion, and Conclusions Sections.
Author Response
Response to Reviewer 3
Thank you to the reviewer for additional suggestions.
We have revised the Material & Methods section as follows to avoid further misunderstanding of the existing colonies.
MCF-7 breast cancer wild type cells were exposed to increasing concentrations of PTX (5-100 nM) over a period of 6 months to generate repetitive clones of PTX-res cells against 100 nM PTX treatment. The generation of PTX-resistant MCF-7 cells was performed in at least three different experiments as different batch clones. In order to prevent the loss of PTX resistant colonies, we prepared 6 different batch clones and utilized at least three of them in confirmation experiments. Cells were treated with PTX for 24 hours. Then, cell debris was removed and remained cell population was maintained in fresh media for at least 1 week. The increased PTX concentration was shown in Figure 1A.
We then searched and added further information. Other cell lines showed similar transcriptomic results according to bio project data.
In addition to several bio project data, which reflect the variations between experiments, our manuscript underlines the importance of JNK, and Wnt related signaling axis may be crucial to overcoming drug resistance. According to GSE12791 while FOS expression levels were increasing in PTX resistant MDA-MB-231 cells, Wnt1 inhibitor DKK and Cadherin were downregulating (Figure 1a-c). Our findings prove the potential active cellular targets in differential responses in MCF-7 cells. This information was also added to the discussion part.
The second issue on the comment about "invasiveness" and "metastasis" was also revised in the manuscript.
We are thankful for the valuable comments
Best regards

Round 3
Reviewer 1 Report
The new response to my concerns is more satisfactory. I think that in this new version of the article can be published in BiologyReviewer 3 Report
The authors have addressed most of the concerns raised.
The manuscript can be considered acceptable for publication once the quality of the English will be fixed.
I pinpointed many mistakes in the correct use of English in almost every new sentence added and highlighted in yellow. Please, re-check the text several times.